# High-Performance Silicon Nanowire Array Biosensor for Combined Detection of Colorectal Cancer Biomarkers

**DOI:** 10.3390/mi16101089

**Published:** 2025-09-26

**Authors:** Jiaye Zeng, Mingbin Liu, Xin Chen, Jintao Yi, Wenhe Liu, Xinjian Qu, Chaoran Liu, Serestina Viriri, Guangguang Yang, Xun Yang, Weichao Yang

**Affiliations:** 1School of Electronic and Information Engineering, China West Normal University, Nanchong 637002, China; 18783008545@163.com (J.Z.); liumb926@163.com (M.L.); cxing202303@163.com (X.C.); 17720707562@163.com (J.Y.); 17645482284@163.com (W.L.); 17669478062@163.com (X.Q.); 2Ministry of Education Engineering Research Center of Smart Microsensors and Microsystems, College of Electronics and Information, Hangzhou Dianzi University, Hangzhou 310018, China; liucr@hdu.edu.cn; 3School of Mathematics, Statistics & Computer Science, University of KwaZulu-Natal, Durban 4041, South Africa; viriris@ukzn.ac.za; 4School of Electronic and Information Engineering, Foshan University, Foshan 528000, China; guangguangyanguop@gmail.com

**Keywords:** silicon nanowire arrays, highly controllable, low cost, combined detection, colorectal cancer biomarkers

## Abstract

This study presents a high-performance silicon nanowire (SiNW) array biosensor for the combined detection of two key colorectal cancer (CRC) biomarkers: circulating tumor DNA (ctDNA) and carcinoembryonic antigen (CEA). The device was fabricated using conventional micromachining techniques, enabling the integration of dual SiNW arrays on a single chip with precise control over structure and surface functionalization. Specific probe DNA and anti-CEA antibodies were immobilized on distinct array regions to facilitate targeted binding. The biosensor demonstrated exceptional performance, achieving an ultralow detection limit of 10 aM for ctDNA with a linear range from 0.1 fM to 10 pM, and a sensitivity of 1 fg/mL for CEA. It exhibited high selectivity against interfering substances, including single-base mismatched DNA and non-specific proteins, and maintained robust performance in human serum samples. The platform offers a scalable, label-free, and real-time detection solution with significant potential for application in early CRC screening and personalized medicine.

## 1. Introduction

Colorectal cancer (CRC) ranks as the third most prevalent malignancy globally, following lung and breast cancers, with newly diagnosed cases surpassing 1.9 million in 2024 [1,2,3,4]. Early detection is critical for improving patient survival rates, yet the current reality remains concerning: up to 50% of primary CRC patients present with or later develop liver metastases at initial diagnosis [5]. Currently, colonoscopy is regarded as the gold standard for clinical detection of colorectal cancer (CRC), as it allows for direct visualization of the intestinal mucosa, detection of early-stage cancers and polyps, and enables biopsy and removal during the procedure. However, its invasive nature, cumbersome preparation (requiring bowel cleansing), high cost, and inherent risks associated with an invasive procedure contribute to generally low screening adherence in the population [6,7]. Among non-invasive alternatives, the fecal occult blood test is low-cost and suitable for large-scale initial screening, but its low sensitivity leads to high miss rates, and it is prone to false positives due to dietary interference [8]. Therefore, it is imperative to develop more cost-effective, highly sensitive, and widely acceptable detection technologies.

Liquid biopsy (LB) is a minimally invasive diagnostic strategy used for cancer screening and detection [9]. With this technology, early screening and dynamic monitoring of CRC can be achieved through combined detection of circulating tumor DNA (ctDNA) and carcinoembryonic antigen (CEA) [10,11,12,13,14,15,16,17,18,19]. ctDNA carries tumor-specific genetic mutations, providing real-time molecular insights into tumor characteristics [20,21,22,23], while CEA plays a crucial role in assessing therapeutic efficacy and monitoring recurrence risk [24,25,26]. Notably, while single-marker assays are often compromised by individual variability and tumor heterogeneity, recent studies demonstrate that multiplexed detection of multiple biomarkers significantly improves diagnostic accuracy and enables more precise tracking of disease progression [27,28].

Currently, multiple detection techniques for CEA have been developed, including radioimmunoassay, fluorescence labeling, enzyme-linked immunosorbent assay (ELISA), and immunochemiluminometric assay [29]. While these methods enable the detection of CEA, they often suffer from limitations such as complex instrumentation, high cost, time-consuming procedures, and generally insufficient sensitivity. Additionally, radioimmunoassay have been gradually phased out due to radioactive hazards. Some novel fluorescence sensing techniques show significant potential, though they still exhibit certain limitations. Miao H et al. [30]. developed a label-free fluorescent sensing strategy based on carbon dots, which achieved highly sensitive detection of CEA with a limit of detection as low as 0.3 ng/mL. However, the method’s reliance on specific reaction conditions and its complex procedures limit its broad application. Similarly, multiple methods exist for detecting ctDNA, such as quantitative polymerase chain reaction (qPCR) and digital PCR; however, these approaches rely on expensive equipment and intricate multi-step procedures, which poses challenges for large-scale application [31]. Moreover, some conventional techniques suffer from a major limitation: a limited detection range. Garm K S et al. [32]. established a technology platform based on ARMS-qPCR (Amplification Refractory Mutation System quantitative PCR) for ctDNA detection. However, this method is limited to detecting specific known mutations and cannot identify novel or unknown variants, which significantly restricts its utility in samples with high tumor heterogeneity or rare mutations. Therefore, it is imperative to develop novel detection techniques for ctDNA and CEA that are highly sensitive, cost-effective, and easy-to-operate in order to overcome these limitations.

In recent years, nanomaterial-based biosensors have demonstrated remarkable potential in cancer diagnostics. This is due to their unique physicochemical properties, such as high surface-to-volume ratio, quantum confinement effects, and superior biocompatibility. These properties mean that nanostructured sensing platforms can significantly enhance detection sensitivity, specificity, and multiplexing capability. However, key challenges remain in their clinical translation, particularly in achieving scalable, reproducible fabrication of nanostructures and precise control over surface functionalization. For instance, gold nanoparticles and carbon nanotubes enable ultrasensitive biomarker detection but face limitations due to complex synthesis protocols and potential biosafety concerns, which restricts their practical deployment [33,34]. Jandas P et al. [35]. achieved specific recognition of CEA with a detection limit of 0.084 ng/mL using nanocomposites as a transducer interface; however, the complex preparation process involving multi-step nanomaterial synthesis and polymer integration hinders large-scale application. Chai H et al. [36]. constructed an electrochemical biosensor based on DNA triangular prism (TP) and three-way junction (TWJ) nanostructures for ultrasensitive ctDNA detection, but the method suffers from operational complexity and limited adaptability in real samples. Therefore, there is an urgent need to develop novel nanomaterials to address the limitations of conventional materials in sensing applications.

Silicon nanowire (SiNW) sensors show great potential in cancer biomarker detection due to their ultra-high sensitivity, specificity, and real-time monitoring capability [37]. SiNWs are typically fabricated using methods such as chemical vapor deposition (CVD), metal-assisted chemical etching (MACE), and top-down nanofabrication techniques. The first two approaches are often limited by poor process controllability, low uniformity, and potential metal contamination. While top-down strategies offer advantages in controllability and reproducibility, most of them rely on expensive lithographic equipment [38,39,40]. To address these challenges, this work presents an integrated biosensing platform based on SiNW arrays for the simultaneous detection of colorectal cancer biomarkers—circulating tumor DNA (ctDNA) and carcinoembryonic antigen (CEA). This device fulfills the aforementioned requirements for high sensitivity, low cost, high controllability, label-free detection, and operational convenience. It is fabricated using conventional micro-fabrication techniques, requiring only standard photolithography, reactive ion etching, and wet etching [41]. The design supports parallel integration of two functionally customized SiNW arrays on a single chip and enables precise control over surface chemistry. Our sensor demonstrates a detection limit of 10 aM for ctDNA (linear range: 0.1 fM to 10 pM) and 1 fg/mL for CEA (linear range: 1 fg/mL to 10 pg/mL), outperforming previously reported biosensors in detection performance while maintaining high selectivity. By enabling multiplexed, highly sensitive, and selective biomarker detection, this SiNW system provides a scalable and cost-effective solution for the early diagnosis of colorectal cancer.

## 2. Materials and Methods

### 2.1. The Fabrication of SiNW Array Biosensor

As illustrated in Figure 1, the fabrication process begins with a boron-doped (111)-oriented silicon-on-insulator (SOI) substrate, the SOI structure comprises a 20-μm device layer, 500 nm buried oxide, and 380-μm handle layer, fabricated via smart cut technology for precise thickness control. Initial processing involves depositing a 100 nm low-stress silicon nitride (Si_3_N_4_) film via low-pressure chemical vapor deposition at 780 °C—optimized for stoichiometric Si_3_N_4_ formation and minimal intrinsic stress. As shown in Figure 1a, the photolithography is then used to pattern an array of tilted rectangular windows in the nitride film. A dry-etch process selectively removes the exposed silicon nitride and underlying silicon within these windows, creating 20-μm-deep rectangular trenches (Figure 1b). Subsequent anisotropic wet etching in potassium hydroxide solution exploits the markedly slower etching rate of the (111) crystal plane, progressively transforming the trenches into hexagonal grooves bounded by (111) sidewalls. This process creates inclined thin-silicon membranes between adjacent grooves (Figure 1c). A self-limiting thermal oxidation step follows, the presence of Si_3_N_4_ at the crest of the silicon walls retards oxide formation in these regions, preserving unoxidized silicon cores that constitute the SiNW array (Figure 1d). Electrode fabrication utilizes ion implantation for p-doped contact regions and magnetron sputtering of 500 nm gold (Au) pads with titanium (Ti) adhesion layers. Finally, deep reactive ion etching creates isolation trenches, providing physical isolation between anode/cathode regions to minimize leakage currents (Figure 1e). Subsequently, buffered oxide etchant is used to remove the silicon oxide walls, which can expose the SiNWs for subsequent surface modification (Figure 1f).

As shown in Figure 2, two arrays were designed and fabricated on a single chip to facilitate subsequent multiplex detection of colorectal cancer biomarkers. After removing the top silicon nitride layer, fully released SiNW arrays were obtained, with both ends of the nanowires monolithically integrated into the bulk silicon, enabling direct electrical connections without requiring transfer processes. Through this straightforward microfabrication process, we have fabricated a highly controllable, miniaturized and low-cost SiNW array sensor for colorectal cancer marker detection.

### 2.2. The Reagents and Test Equipment

The reagents used in this experiment included N-hydroxysuccinimide (NHS), 1-ethyl-3-(3-dimethylaminopropyl) carbodiimide (EDC), phosphate-buffered saline (1 × PBS, containing 137 mM NaCl, 2.7 mM KCl, and 10 mM phosphate, pH 7.2–7.4), and anhydrous ethanol, which were supplied by Macklin Biochemical Co., Ltd. (Shanghai, China). 3-Aminopropyltriethoxysilane (APTES), glutaraldehyde, ethanolamine, prostate-specific antigen (PSA), and bovine serum albumin (BSA) were purchased from Sigma-Aldrich (St. Louis, MO, USA). Carcinoembryonic antigen (CEA) and its corresponding antibody were obtained from Fitzgerald Inc. (Elkader, IA, USA). All chemicals were handled strictly in accordance with the manufacturers’ instructions. Furthermore, ctDNA-related oligonucleotides were synthesized by Sangon Biotech Co., Ltd. (Shanghai, China). The DNA probes used included an ssDNA probe, complementary DNA (ctDNA), one-base mismatched DNA, two-base mismatched DNA, and non-complementary DNA, with the following respective sequences: 5′-COOH-CCCCCCAGTGATTTTAGAGAG-3′, 5′-CTCTCTAAAATCACT-3′, 5′-CTCTCTGAAATCACT-3′, 5′-CTCTCTGAAATCACT-3′ and 5′-TACTCCGCGCTAACG-3′.

Electrical characterization of the SiNW array sensor, including sensitivity, specificity, and linear concentration range, was performed using a Keithley 2450 semiconductor parameter analyzer, this device was manufactured by Keithley Instruments, Inc. (a subsidiary of Tektronix, Inc.), Cleveland, OH, USA. All measurements were conducted at ambient temperature.

### 2.3. The Surface Modification of the SiNW Array Biosensor

The surface modification process for the multiplexed SiNW array biosensor is illustrated in Figure 3, we have made different modifications on the two arrays to achieve different detection functions. Initially, the sensor undergoes oxygen plasma treatment at 70 W power to clean the nanowire array surfaces while simultaneously generating abundant hydroxyl groups, thereby enhancing surface hydrophilicity. Subsequently, the SiNW array sensor was immersed overnight in a 2% APTES solution diluted with 99% ethanol for surface modification. The two sensing regions of the sensor were then functionalized separately. On the first sensing region of the sensor, 6 μL of 10% glutaraldehyde solution was applied and incubated in the dark for 2 h. Subsequently, 6 μL of CEA antibody solution (2.033 mg/mL) was added, followed by 6 μL of ethanolamine solution (9.9%). The ethanolamine reacts with unreacted aldehyde groups to minimize nonspecific binding currents arising from interactions between interfering substances and residual active sites. For the other sensing region, the terminal carboxyl groups of the probe DNA were activated prior to immobilization by incubating with an equimolar mixture of NHS and EDC for 30 min. Subsequently, a 0.5 μM single-stranded DNA probe in PBS solution was introduced into the sample chamber and incubated at room temperature for 2 h. Finally, the surface was rinsed with PBS to remove unbound probe molecules. Following the immobilization steps, a recognition functional layer was formed on the sensor surface, indicating that the specific biosensor was ready for target ctDNA detection. Through these procedures, an optimized SiNW array biosensor was fabricated for simultaneous detection of ctDNA and CEA. Figure 3b illustrates the sensing mechanism of the sensor, when negatively measured object (ctDNA and CEA) are specifically adsorbed onto the nanowire surface, the hole concentration within the P-type nanowire channel increases, thereby leading to an enhancement in the current signal.

## 3. Results

### 3.1. The Detection and Sensitivity of the SiNW Array Biosensor

As mentioned previously, the SiNW array sensor enables simultaneous detection of dual biomarkers—using surface-immobilized DNA probes and CEA antibodies—with its core mechanism relying on the conductive properties of the p-type SiNWs. To achieve high-sensitivity and high-selectivity detection of ctDNA and CEA, DNA probes and CEA antibodies were immobilized on distinct sensing regions of the sensor. Upon introduction of solutions containing 10 aM ctDNA or 1 fg/mL CEA, the current increased rapidly within minutes (Figure 4a,b), indicating that the binding of target molecules to the modified probes or antibodies induced a significant increase in the surface charge density of the SiNWs, thereby leading to a rise in current. Furthermore, by testing a series of concentrations of ctDNA (0.1 fM–10 pM) and CEA (1 fg/mL–10 pg/mL), the sensor was verified to possess a broad linear detection range (Figure 4c,d). To demonstrate the excellent linearity of our device’s response, we performed a linear regression analysis on the data across a wide concentration range and determined the coefficient of determination (R^2^).Y = a + b × X(1)
where a is the intercept, b is the slope, and X represents the logarithm of the ctDNA or CEA concentration in this study. Following calibration and fitting, the linear regression equations for ctDNA and CEA were determined to be Y = 1.40044 × 10^−6^ + 1.47035 × 10^−7^ × X (R^2^ = 0.99469) and Y = 1.12461 × 10^−6^ + 5.44623 × 10^−7^ × X (R^2^ = 0.99919). The results demonstrate a highly linear dependence between the relative change in current and the logarithm of the target analyte concentration. We calculated the limit of detection (LOD) and compared it with the threshold current [42].Y_LOD_ = Y_blank_ + 3σ(2)
where Y_blank_ denotes the blank response and σ represents its standard deviation. For ctDNA and CEA, the Y_blank_ values were 1.40362 × 10^−^^6^ A and 1.22523 × 10^−^^6^ A, with σ values of 2.80618 × 10^−^^9^ A and 4.9515 × 10^−^^9^ A, respectively. The corresponding threshold currents were calculated to be 1.41204 × 10^−^^6^ A and 1.24008 × 10^−^^6^ A. Under exposure to 10 aM ctDNA and 1 fg/mL CEA, the steady-state current responses reached 1.50207 × 10^−^^6^ A and 1.57151 × 10^−^^6^ A, respectively, both exceeding their respective thresholds. Therefore, the sensor reliably detected ctDNA at concentrations as low as 10 aM and CEA at 1 fg/mL. These results demonstrate that the fabricated SiNW array sensor exhibits remarkably high sensitivity towards both ctDNA and CEA, enables rapid detection, and is suitable for applications such as intraoperative real-time monitoring.

### 3.2. The Specificity of the SiNW Array Biosensor

To evaluate the specificity of the sensor for target ctDNA and CEA recognition, experiments were designed to distinguish single-base mismatches in ctDNA and to assess anti-interference capability against CEA. In blood, ctDNA typically exists as short fragments ranging from 50 to 250 base pairs in length [43]. Among these, the E542K site in the PIK3CA gene is a common mutation hotspot, often characterized by an A-to-G substitution. To evaluate the sensor’s ability to discriminate mutant sequences, tests were conducted using 10 pM of one-base mismatched DNA (simulating the E542K mutation), two-base mismatched DNA, and non-complementary DNA. The results demonstrated that both non-complementary and two-base mismatched DNA induced minimal current changes (~−3.2% and ~15%, respectively), indicating no significant response. In contrast, the one-base mismatched DNA elicited a 31% current change, while the complementary ctDNA produced the highest response signal at 102% (Figure 5a). The unmodified sensor showed no obvious response to 10 pM ctDNA (Figure 5b). These findings indicate that the sensor can accurately distinguish among complementary, single-base mismatched, double-base mismatched, and non-complementary DNA sequences, demonstrating high specificity and potential applicability for screening colorectal cancer-related gene mutations. Subsequently, in the CEA antibody-modified sensing region, high-concentration interferents (10 mg/mL BSA and 100 μg/mL PSA) followed by 1 fg/mL CEA were introduced sequentially. The results showed that the current change induced by the interferents was negligible, whereas a significant current response of 35% was triggered by 1 fg/mL CEA (Figure 5c). No response was observed with the unmodified sensor toward 10 pg/mL CEA (Figure 5d), confirming that the detection specificity originates from the specific antibody–antigen binding. These findings demonstrate that the sensor, through precise probe/antibody modification and charge amplification effects, achieves highly specific detection of both CEA and ctDNA. It enables effective discrimination of single-base mutations, specific recognition of antibody–protein interactions, and resistance to interference in complex sample matrices, thereby providing key technological support for the precision diagnosis of colorectal cancer.

### 3.3. The Detection in Human Serum Samples

To evaluate the sensor’s performance in real biological samples, this study investigated its detection capability for CEA in a serum sample. CEA-containing serum samples were subjected to gradient dilution using PBS buffer. As illustrated in Figure 6, upon addition of the diluted CEA serum solution, the current increased rapidly. The results demonstrate that the SiNW biosensor achieved a detection sensitivity as low as 10 fg/mL for CEA in serum. As the CEA concentration increased from 10 fg/mL to 100 pg/mL, the response current exhibited a distinct step-wise enhancement. Corresponding to CEA concentrations of 10 fg/mL, 100 fg/mL, 1 pg/mL, 10 pg/mL, and 100 pg/mL, the current increased by 83%, 132%, 202%, 258%, and 317%, respectively. The sensing system reached response equilibrium within 120 s and showed excellent linearity (R^2^ = 0.99769) between current change and analyte concentration. These results confirm that the sensor maintains high sensitivity and reliability even in complex matrices such as serum.

Currently, conventional detection methods for cancer biomarkers generally suffer from insufficient sensitivity, long detection times, and reliance on large, expensive equipment. Table 1 and Table 2 compare the performance of the developed SiNW array biosensor with several conventional sensors in terms of limit of detection (LOD), test range, and response time. The results indicate that existing methods struggle to achieve high-sensitivity detection of target cancer biomarkers. Although some technologies offer high sensitivity, they are constrained by complex device fabrication processes and multi-step detection procedures, limiting their practical application.

The SiNW array biosensing technology proposed in this study effectively overcomes these limitations. This technology offers advantages such as label-free operation, electrical readout, high sensitivity, low cost, and real-time monitoring, demonstrating strong potential for early cancer diagnosis. Compared to other nanomaterial-based devices, the SiNW array provides stronger signal response and higher signal-to-noise ratio, and can be easily integrated with low-cost electronic detection systems. This enables the development of a cost-effective detection solution suitable for early screening of colorectal cancer.

## 4. Conclusions

In summary, this study successfully developed a multifunctional dual-mode biosensor based on uniform SiNW arrays for the simultaneous detection of colorectal cancer biomarkers, ctDNA and CEA. The sensor was fabricated using a wafer-scale CMOS-compatible process. Through precise modification with probe DNA and CEA antibodies, a dual-mode detection platform was established. Using the CMOS compatibility of the fabrication process, the biosensor allows low-cost, highly controllable, and large-scale production. Experimental results demonstrate excellent specificity, sensitivity, repeatability, and reproducibility. The sensor achieves an ultra-low detection limit of 10 aM for ctDNA, with a broad linear range from 0.1 fM to 10 pM. For CEA, the detection limit is as low as 1 fg/mL, with a linear range from 1 fg/mL to 10 pg/mL. It accurately distinguishes single-base mismatches and specifically recognizes target proteins, even in complex serum environments, significantly improving detection efficiency compared to conventional methods. This technology provides an innovative tool for real-time monitoring and personalized treatment of colorectal cancer. Future integration into miniaturized designs may enable its application in portable medical devices, promoting the widespread adoption of precision cancer diagnostics.

## Figures and Tables

**Figure 1 micromachines-16-01089-f001:**
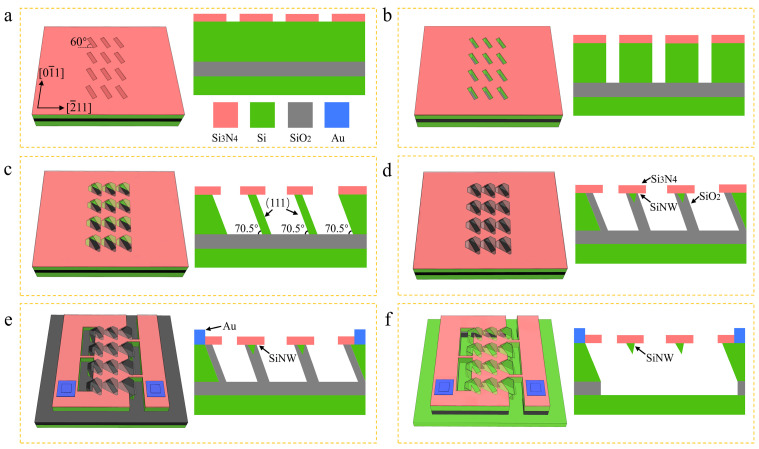
The fabrication process of SiNW array sensor. (**a**) Photolithography patterning and reactive ion etching for window array formation, left panel shows a cross-sectional schematic; (**b**) groove etching via deep reactive ion etching; (**c**) formation of tilted silicon sidewalls by anisotropic wet etching; (**d**) self-limiting thermal oxidation for SiNW array formation; (**e**) electrode deposition and isolation channel patterning; (**f**) oxide removal to expose the SiNWs.

**Figure 2 micromachines-16-01089-f002:**
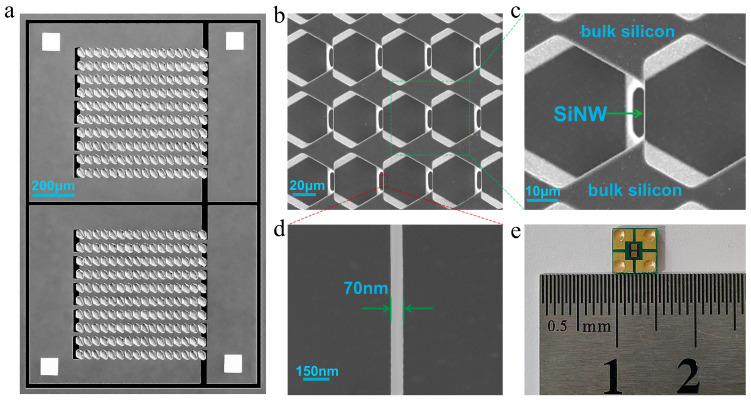
(**a**) The electron microscopy image of SiNW array biochip; (**b**) the fully released SiNW array; (**c**) the localized magnification of SiNW array; (**d**) the localized magnification of SiNW; (**e**) the packaged SiNW array biochip.

**Figure 3 micromachines-16-01089-f003:**
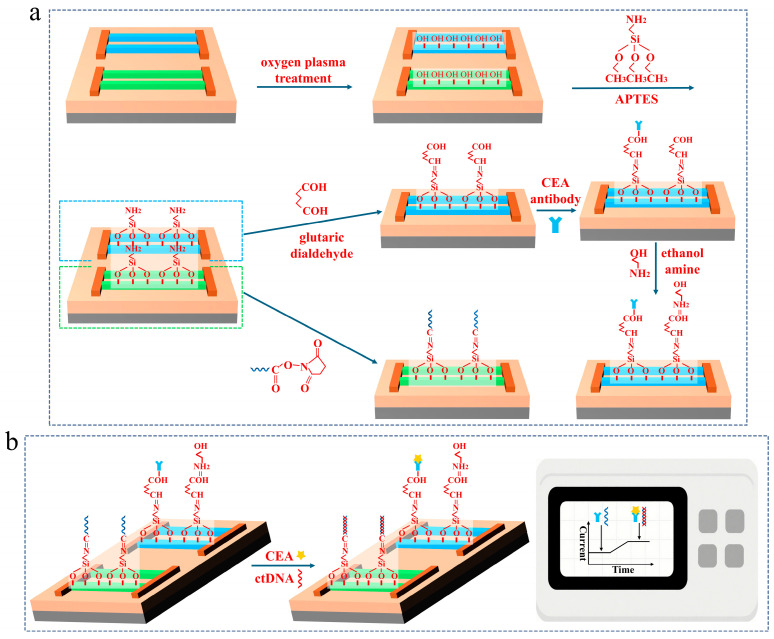
(**a**) The pre-functionalization treatment of the SiNW array biosensor; (**b**) the detection principle of SiNW array biosensor.

**Figure 4 micromachines-16-01089-f004:**
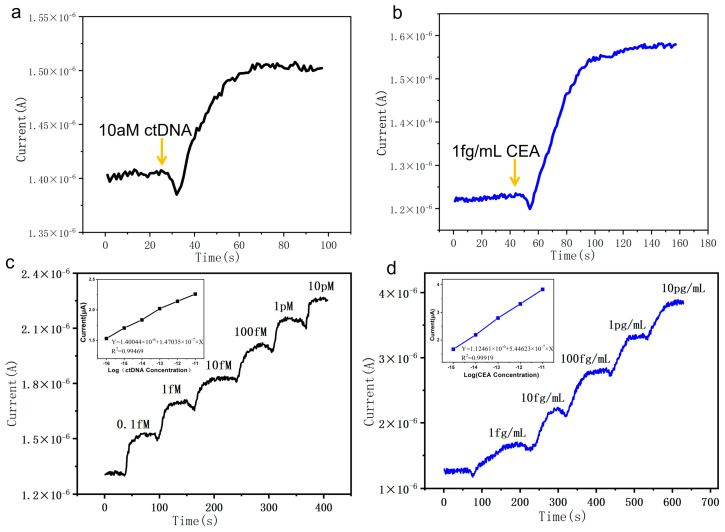
(**a**) Real-time current response of the silicon SiNW array biosensor to 10 aM ctDNA; (**b**) response to 1 fg/mL CEA; (**c**) responses to ctDNA at concentrations of 0.1 fM to 10 pM; (**d**) responses to CEA at concentrations ranging from 1 fg/mL to 10 pg/mL.

**Figure 5 micromachines-16-01089-f005:**
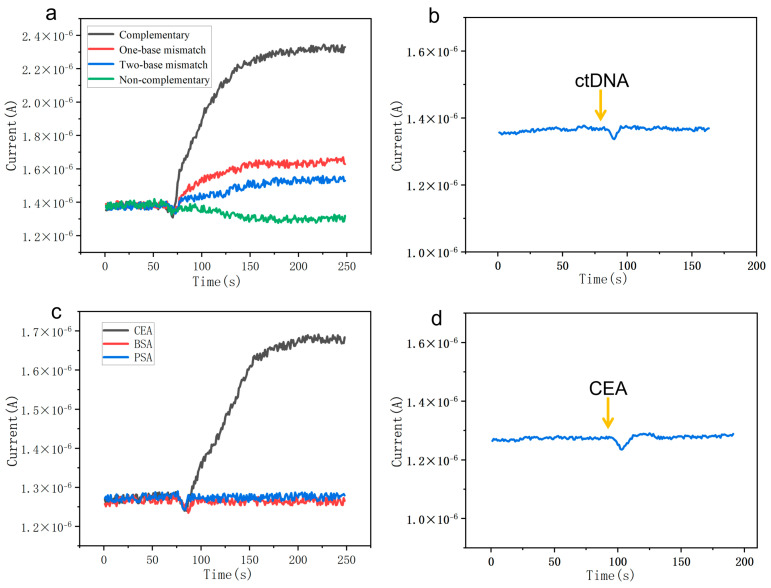
(**a**) Current responses under different hybridization conditions at the same sample concentration (10 pM); (**b**) real-time current response of an unmodified SiNW array biosensor to 10 pM ctDNA; (**c**) real-time current change in the SiNW in response to 1 fg/mL CEA, 10 mg/mL BSA, and 100 µg/mL PSA; (**d**) real-time current response of an unmodified SiNW array biosensor to 10 pg/mL CEA.

**Figure 6 micromachines-16-01089-f006:**
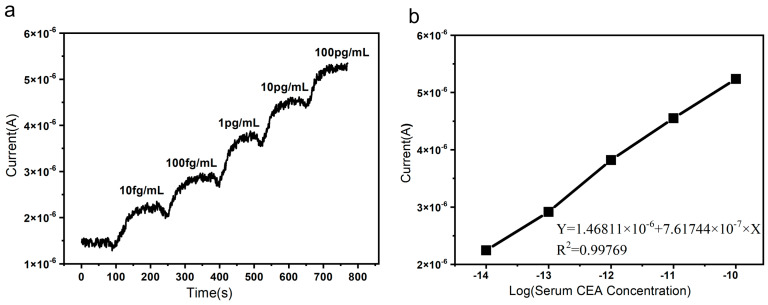
(**a**) Real-time response of the SiNW array biosensor to different concentrations of CEA in serum (ranging from 10 fg/mL to 100 pg/mL); (**b**) relative current change in the SiNW array biosensor as a function of the logarithm of CEA concentration in serum.

**Table 1 micromachines-16-01089-t001:** Comparison of detection limit and test range for ctDNA.

Methods	Detection Limit (μM)	Test Range (μM)	References
SERS frequency shift assay	9.0 × 10^–9^	10^–9^~10^–2^	[44]
Nucleic acid biosensor	2 × 10^−6^	5 × 10^−3^~1	[45]
NIR fluorescent nanoprobes	6.3 × 10^−6^	5 × 10^−6^~10^−3^	[46]
Electrochemical biosensor	3 × 10^–6^	5 × 10^–4^~5 × 10^–2^	[47]
Fluorescent biosensor	5 × 10^−8^	5 × 10^−8^~8×10^−5^	[48]
SiNW array biosensor	10^−11^	10^−10^~10^−5^	This work

**Table 2 micromachines-16-01089-t002:** Comparison of detection limit and test range for CEA.

Methods	Detection Limit (pg/mL)	Test Range (pg/mL)	References
Surface plasmon resonance	120	400~20,000	[49]
Chemiluminescence immunoassay	610	610~250,000	[50]
Click Chemistry-Based ELISA	540~660	0~200,000	[51]
Fluorescent biosensor	210	210~200,000	[52]
Electrochemical Immunosensor	0.286	1~80,000	[53]
SiNW array biosensor	0.001	0.001~10	This work

## Data Availability

The original contributions presented in the study are included in the article, further inquiries can be directed to the corresponding author.

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
