# Peer review of "High-Performance Silicon Nanowire Array Biosensor for Combined Detection of Colorectal Cancer Biomarkers"

_micromachines, 2025, doi:10.3390/mi16101089_

Round 1

Reviewer 1 Report

Comments and Suggestions for Authors

In the manuscript, the authors present a design of Si nanowire (SiNW) arrays on a single chip as a biosensor, The sensor realized multiplexed detection of ctDNA and CEA with high selectivity and sensitivity, and specifically recognized target proteins in complex serum environments. The work as a whole is engaging and shows great promise in terms of potential applications. However, the following issues must be clarified before a decision to accept the proposal can be made.
1. As shown in Fig. 1(a), the SiNWs were created with a specific titling angle. How to control the angle? It is suggested to supply detailed experimental parameters.
2.  The figure number of each step in Figure 1 should be supplied in the description of the fabrication process.
3. It is not Figure 2b but 3b in line 175.
4. The figure sequence in Figure 5 needs to be re-organized in light of the discussion. 
5. The figure, table and video in the supplementary materials were not cited in the manuscript. What role do they play?
6. Please provide a table for comparing detecting range, sensitivity, etc of the SiNW arrays with other biosensors.

Reviewer 2 Report

Comments and Suggestions for Authors

The manuscript presents a fabricated silicon nanowire (SiNW) array biosensor for multiplex detection of circulating tumor DNA (ctDNA) and carcinoembryonic antigen (CEA). The experimental results are promising, and the demonstration of an operational device adds credibility and relevance to the study. To enhance clarity and impact, the paper would benefit from a tighter narrative and a more rigorous technical presentation. In particular, the exposition should guide the reader from the clinical motivation to the sensor design, and then to fabrication, with a brief design rationale introduced before the process flow to justify key architectural choices.

Beyond structure, the manuscript would gain from greater technical depth and a more consistently scientific register. A quantitative comparison with recent SiNW-based biosensors, reporting parameters (LOD/linear range, response time, sensitivity) would help situate the contribution within the state of the art and make the novelty claims more transparent. Finally, a careful language revision is recommended to improve precision, terminology consistency, and formatting. These suggestions are offered constructively: the core results are solid, and with these refinements the work can be communicated more clearly and persuasively.

Here, my comments:

  • The abstract requires a substantial revision for language and clarity.
  • The sentence “Liquid biopsy is regarded as a promising non-invasive approach for CRC diagnosis” (line 39-40) is too narrow and not fully accurate. Liquid biopsy (LB) is a minimally invasive, diagnostic and monitoring strategy that includes, but is not limited to, colorectal cancer. Please broaden the framing beyond CRC and use “minimally invasive” rather than “non-invasive.” Add a brief contextualization of LB to justify the biosensor design (see, e.g., recent review paper as Shining the Path of Precision Diagnostic: Advancements in Photonic Sensors for Liquid Biopsy. Biosensors, 15(8), 473, 2025).
  • The Introduction requires a substantive overhaul. Although the statements are broadly correct, they read as generic and lack the technical depth needed to motivate the study. Provide a balanced appraisal of conventional assays, CEA (RIA, ELISA/CLIA) and ctDNA (qPCR/dPCR/NGS), covering both advantages and limitations with quantitative metrics (typical LOD/linear range, time-to-result, throughput, instrument complexity/cost). Clarify the biosensor technology landscape, emphasizing performance parameters (LOD, FOM, bulk sensitivity, surface sensitivity, etc) and translation barriers. Close with a precise gap statement that justifies the SiNW array choice versus alternatives and explains how your approach addresses known pain points. Consider a compact SoA table summarizing recent CEA/ctDNA sensors main performance parameters.
  • Line 155. Replace “We” with “we” (remove the leading space and use lowercase, as it appears mid-sentence).
  • Line 175. Typo in the figure callout: it should read “Fig. 3b”. Please verify the entire manuscript for consistent and correct numbering/cross-references of all figures and tables (captions, in-text callouts, and reference list).
  • In line 189, please define the term “ISD”.
  • From line 194 onward, please ensure strict compliance with the journal template and style guide, with particular attention to equations.
  • The results are promising and technically significant. To properly contextualize the contribution and clarify its positioning within the literature, a quantitative benchmark against recent silicon nanowire (SiNW) array biosensors is recommended. Please include a concise comparative analysis (preferably a table) for both ctDNA and CEA sensing’s.

Round 2

Reviewer 1 Report

Comments and Suggestions for Authors

The authors have resolved the problems as I listed last time.

Reviewer 2 Report

Comments and Suggestions for Authors

The Authors have modified the manuscript according to the Reviewers' suggestions.